# SwitchLoss: A Novel Optimization Scheme for Imbalanced Regression

## Abstract

In the realm of machine learning, conventional techniques like neural networks often encounter challenges when dealing with imbalanced data. Unfortunately, imbalanced data is a common occurrence in real-world datasets, where collection methods may fail to capture sufficient data within specific target variable ranges. Additionally, certain tasks inherently involve imbalanced data, where the occurrences of normal events significantly outweigh those of edge cases. While the problem of imbalanced data has been extensively studied in the context of classification, only a limited number of methods have been proposed for regression tasks. Furthermore, the existing methods often yield suboptimal performance when applied to high-dimensional data, and the domain of imbalanced high-dimensional regression remains relatively unexplored. In response to the identified challenge, this paper presents SwitchLoss, a novel optimization scheme for neural networks, and SwitchLossR, a variant with a restricted search space. Diverging from conventional approaches, SwitchLoss and SwitchLossR integrate variable loss functions into the traditional training process. Our assessment of these methods spans 15 regression datasets across diverse imbalanced domains, 5 synthetic high-dimensional imbalanced datasets, and two imbalanced age estimation image datasets. Findings from our investigation demonstrate that the combined utilization of SwitchLoss and SwitchLossR not only leads to a notable reduction in validation error, but also surpasses prevailing state-of-the-art techniques dedicated to imbalanced regression.

## 1 Introduction

In recent years, the growing availability of large datasets has enabled researchers to apply machine learning for building predictive models. However, many real-world datasets exhibit imbalanced or skewed distributions, which can hinder model performance, particularly in regions with sparse data. While imbalanced classification has received attention, imbalanced regression remains less explored. Only a few methods address imbalanced regression, most focusing on sampling techniques. Notable examples include SMOGN (Branco et al., 2017), an enhancement of SMOTER using Gaussian noise (Torgo et al., 2013). Another example is DenseLoss (Steininger et al., 2021), a cost-sensitive method designed for imbalanced regression, that avoids the removal of potentially useful data by focusing on optimization challenges.

In response to this challenge, we propose a novel optimization scheme to address the problem of imbalanced regression. This method, which we named SwitchLoss, introduces a novel approach that leverages the dynamic switching between different loss functions during the training process of neural networks, thereby acting as a regularization technique and mitigating the risk of optimization converging to a local optimum.

This paper is organized as follows: Section 2 defines the problem of imbalanced regression and presents an overview of the existing related work. The proposed optimization scheme SwitchLoss is described in Section 3, while the results of the experimental evaluation are presented in Section 4. We provide a discussion on the obtained results in Section 5. Finally, Section 6 outlines the main conclusions and future work.

## 2 RELATED WORK

Imbalanced regression refers to regression problems where the target variable is unevenly distributed, with some ranges being underrepresented. The goal is to build models that accurately approximate the function $Y = f(x)$ using a training set $D = (x_i, y_i)_{i=1}^{N}$ with $N$ samples. Unlike classification, imbalanced regression is more complex due to the continuous nature of the target variable.

A significant portion of existing approaches builds upon the seminal work of Torgo and Ribeiro (2007) and Ribeiro (2011), who proposed the concept of a relevance function $\phi(y)$ which assigns a quantitative score [0,1] to the range of target values. Furthermore, Ribeiro (2011) introduced an automated method for approximating the relevance function $\phi(y)$ using box plot statistics. This approximation assumes that the rare and extreme cases hold the highest relevance. Subsequently, the relevance function is utilized to classify data samples into major (normal) and minor (rare) categories, employing a user-defined threshold $tr$. This division is accomplished through the following assignment: $RS = \{(x, y) \in D : \phi(y) \geq tr\}$ and $NS = \{(x, y) \in D : \phi(y) < tr\}$. This categorization based on the relevance function and the user-specified threshold enables the segregation of the data set into rare and normal instances for further analysis and modeling.

Approaches for handling imbalanced data include resampling and cost-sensitive learning (Krawczyk, 2016). A few different sampling approaches for imbalanced regression are proposed. They are applied during data pre-processing, such as SMOTER (Torgo et al., 2013) which is based on the original SMOTE method for classification (Chawla et al., 2002) and combines under-sampling of common data samples with over-sampling of rare cases, in order to create a more balanced distribution. The SMOGN (Branco et al., 2017) can be considered state-of-the-art among resampling techniques. This algorithm builds on top of SMOTER and combines it with oversampling via Gaussian noise. Normally distributed noise is added to the features and the target value of rare data samples, therefore creating additional, slightly altered replicas of existing samples (Branco et al., 2016). Cost-sensitive methods are far less common in regression, but DenseLoss (Steininger et al., 2021) offers a promising approach. It uses DenseWeight, a density-based weighting scheme, to determine rarity without modifying the dataset.

## 3 METHODS

In this section, we introduce SwitchLoss as a versatile optimization scheme for neural networks and provide a specific outline of its implementation for imbalanced regression problems.

### 3.1 GENERAL SWITCHLOSS SCHEME

In machine learning, optimizing the loss function aims to minimize the difference between a neural network's predicted outputs and the actual outputs, thereby improving model performance. The choice of the loss function reflects the network's objectives, guiding it toward a more accurate input-output relationship as training progresses (Goodfellow et al., 2016). Many machine learning problems involve multiple factors that are difficult to combine in a single loss function due to differences in scales, units etc. In this study, we explore an alternative approach where different loss functions are switched during the training phase, rather than merging them into one.

This approach is highly generalizable and can be used across different domains. We recognize that a machine learning model's success largely depends on its optimization process. To address this, we propose a nested two-stage optimization framework. The first stage, called the exploration stage, optimizes the scheme of loss functions for training. The second stage, traditional training, then optimizes the neural network's parameters using the selected scheme. The traditional training is occurring within the broader scope of the exploration stage. Thus, the first stage evaluates different loss schemes, while the second focuses on optimizing the model based on the chosen scheme.

The proposed approach involves randomly switching between predefined loss functions during specific epochs over a fixed number of cycles. This method explores the effect of different loss functions on training, aiming to identify the most effective configuration. By introducing random switching, the approach increases flexibility and diversity in optimization.

By employing this nested two-stage optimization framework, we aim to enhance the overall performance of the neural network model by effectively optimizing both the loss function scheme and the underlying neural network parameters. The framework's details are outlined in Procedure 1. To demonstrate its feasibility, we implement the scheme in the context of imbalanced regression, with detailed specifics provided in the following subsection.

---

**Procedure 1** General SwitchLoss

---

**Require:** $D$ - $\{(f_i, y_i)\}_{i=1}^N$ data set with $f_i$ feature vectors and $y_i$ continuous target values

$\{Loss f_i\}_1^n$ - set of $n$ loss functions

$explores$ - number of exploration cycles for the optimization scheme

$\#switches$ - number of changes of loss function in traditional training

$epochs$ - number of epochs for the traditional optimization

 

   **procedure** SWITCHLOSS($D$)

      epochs_to_switch = round($\frac{epochs}{\#switches}$)       ▷ Number of epochs with constant loss function

      switch_epochs = $\{$epochs_to_switch $\times$ i$\}_{i=1}^{\#switches-1}$

      loss_function = random($\{Loss f_i\}_1^n$)               ▷ Initialization

      $min\_error = \infty$                           ▷ Initialization

      best_model = Null                     ▷ Initialization

      **for** $e \in (1, explores)$ **do**       ▷ First stage of exploration - different configurations

         **for** $i \in (1, epochs)$ **do**      ▷ Nested traditional training within the exploration stage

            **if** $i \in switch\_epochs$ **then**

               loss_function = random($\{Loss f_i\}_1^n$)

            **end if**

            traditional_training(D)          ▷ With previously defined loss_function

         **end for**

         Calculate $error$ on a test data set          ▷ Compare optimization schemes

         **if** $error < min\_error$ **then**           ▷ Minimum error

            best_model = model         ▷ Save the model with the minimum test error

            $min\_error = error$

         **end if**

      **end for**

      **return** best_model             ▷ Return the model that performs the best

   **end procedure**

---

### 3.2 HYPER-PARAMETERS

Within our algorithm, we acknowledge the presence of three important parameters. The first parameter pertains to the selection of a set of $n$ loss functions, which is a crucial decision requiring domain knowledge and problem-specific considerations. The choice of these functions determines the optimization criteria used during the training process. In the following subsection we propose and justify our choices for the imbalanced regression applications.

The second parameter, denoted as $\#switches$, represents the number of divisions of the traditional training into continuous loss function blocks. The value of $\#switches$ can be adjusted based on the specific characteristics of the problem.

The third parameter, denoted as $explores$, indicates the number of cycles dedicated to exploring various training schemes. Increasing the value of $explores$ enhances the likelihood of identifying the best possible training scheme, while also requiring larger computational resources. Ideally, setting this value to $n^{\#switches}$ would allow for an exhaustive exploration of all available options. In this scenario, random selection becomes inconsequential as the systematic search ensures more meaningful and comprehensive results.

### 3.3 SWITCHLOSS FOR IMBALANCED REGRESSION

In this section, we provide a comprehensive elaboration of the SwitchLoss scheme's implementation for the specific domain of imbalanced regression.

The majority of algorithms employed in the imbalanced regression domain rely on sampling techniques (Krawczyk, 2016). These algorithms typically involve either over-sampling, generating artificial samples, or under-sampling, failing to fully exploit the available information.

In our research, we believe that the suboptimal performance of neural networks in imbalanced regression problems stems from issues in the optimization process. Specifically, we argue that conventional optimization methods excessively prioritize the proportion of data samples within abundant regions. Consequently, this bias towards the abundant regions leads to the attraction of optimization towards local minima, resulting in predictions that predominantly align with values close to the abundant region for any given input. To address this problem, we propose an alternative optimization procedure that leverages the entire available information without resorting to intentionally induced loss or the artificial generation of data samples.

We hypothesize that by introducing a change in the loss functions employed during training, while maintaining the same objective - finding the best approximation of the underlying function, we can effectively steer the optimization away from local minima. Therefore, we propose the utilization of the SwitchLoss scheme, which involves a predefined set of three loss functions in the context of imbalanced regression:

The first choice for a loss function is a standard root-mean-squared error, given in Equation 1.

$$MSE = \frac{1}{N} \sum_{N}^{1} (y - \hat{y})^2 \tag{1}$$

N refers to the number of samples in a training data set, while $y$ and $\hat{y}$ are true target values and predicted target outputs, respectively.

We augment the list of functions by introducing what we refer to as "optimization on the model" instead of "on the data". Our second choice is Jensen-Shannon divergence (JSD), given in Equation 2.

$$JSD(p\|q) = \frac{1}{2} D_{KL}(p\|m) + \frac{1}{2} D_{KL}(m\|q) \tag{2}$$

The JSD is a mathematical measure that assesses the dissimilarity between probability distributions. It quantifies the discrepancy or divergence between two distributions by considering both their similarities and differences (Fuglede & Topsoe, 2009).

For the last function in the optimization procedure we propose the disparity between the standard deviations of the predicted and actual outputs of the neural network, given in Equation 3.

$$STD_{loss} = \|\sigma(y) - \sigma(\hat{y})\| \tag{3}$$

In the equation $\sigma(\hat{y})$ represents the standard deviation of the predicted neural network output, and $\sigma(y)$ denotes the standard deviation of the actual output.

By incorporating this particular function along with JSD into the optimization process, we aim to capture and address discrepancies in the spread of the predicted output compared to the ground truth. Minimizing the absolute difference between these standard deviations encourages the neural network to generate predictions that exhibit similar levels of variability as the actual output, ultimately enhancing the model's ability to accurately capture the underlying distribution's dispersion properties.

The conventional optimization approach encounters challenges in accurately predicting target values within rare regions of the target distribution. As a result, the predicted target values tend to concentrate solely in abundant regions, leading to a narrower predicted standard deviation distribution compared to the actual distribution. By incorporating these loss functions, the model is incentivized to generate predictions that extend beyond the abundant region, thereby encouraging the exploration and accurate prediction of values within rare regions of the target distribution.

Finally, for the application of SwitchLoss to imbalanced regression we propose using the following set of loss functions, given in Equation 4. The details are outlined in Procedure 2.

$$\{Loss f_i\}_1^3 = \{MSE, JSD, STD_{loss}\} \tag{4}$$

---

**Procedure 2** SwitchLoss for Imbalanced Regression

---

**Require:** $D$ - $\{(f_i, y_i)\}_{i=1}^{N}$ data set with $f_i$ feature vectors and $y_i$ continuous target values
$\quad\quad\quad\{MSE, JSD, STD_{loss}\}$ - set of loss functions
$\quad\quad\quad explores$ - number of exploration cycles for the optimization scheme
$\quad\quad\quad \#switches$ - number of changes of loss function in traditional training
$\quad\quad\quad epochs$ - number of epochs for the traditional training

$\quad$**procedure** SWITCHLOSS($D$)
$\quad\quad$epochs_to_switch = round($\frac{epochs}{\#switches}$)$\quad\quad\quad\quad\quad$▷ Number of epochs with constant loss function
$\quad\quad$switch_epochs = $\{$epochs_to_switch $\times$ i$\}_{i=1}^{\#switches-1}$
$\quad\quad$loss_function = random($\{MSE, JSD, STD_{loss}\}$)$\quad\quad\quad\quad\quad\quad\quad$▷ Initialization
$\quad\quad min\_error = \infty$$\quad\quad\quad\quad\quad\quad\quad\quad\quad\quad\quad\quad\quad\quad\quad\quad\quad$▷ Initialization
$\quad\quad$best_model = Null$\quad\quad\quad\quad\quad\quad\quad\quad\quad\quad\quad\quad\quad\quad\quad\quad$▷ Initialization
$\quad\quad$**for** $e \in (1, explores)$ **do**$\quad\quad\quad\quad\quad\quad\quad\quad\quad\quad$▷ First stage of exploration
$\quad\quad\quad$**for** $i \in (1, epochs)$ **do**$\quad\quad\quad$▷ Nested traditional training within the exploration stage
$\quad\quad\quad\quad$**if** $i \in switch\_epochs$ **then**
$\quad\quad\quad\quad\quad$loss_function = random($MSE, JSD, STD_{loss}$)
$\quad\quad\quad\quad$**end if**
$\quad\quad\quad\quad$traditional_training(D)$\quad\quad\quad\quad\quad\quad$▷ With previously defined loss_function
$\quad\quad\quad$**end for**
$\quad\quad\quad$Calculate error on a $balanced$ test data set$\quad\quad\quad\quad$▷ Compare optimization schemes
$\quad\quad\quad$**if** $error < min\_error$ **then**$\quad\quad\quad\quad\quad\quad\quad\quad\quad\quad\quad\quad$▷ Minimum error
$\quad\quad\quad\quad$best_model = model$\quad\quad\quad\quad\quad$▷ Save the model with the minimum test error
$\quad\quad\quad\quad min\_error = error$
$\quad\quad\quad$**end if**
$\quad\quad$**end for**
$\quad\quad$**return** best_model$\quad\quad\quad\quad\quad\quad\quad\quad\quad\quad$▷ Return the model that performs the best
$\quad$**end procedure**

---

### 3.3.1 RESTRICTING THE SEARCH SPACE

A crucial characteristic of this algorithm resides in the extensive array of possibilities related to the combination of loss functions. Specifically, in the context of imbalanced regression, since three functions have been proposed the total number of conceivable schemes amounts to $3^{\#switches}$. As the number of functions or switches increases, this count escalates significantly. In light of limited resources such as time or computational power, we put forth techniques aimed at mitigating the magnitude of the search space.

One of the options is assigning higher probabilities to specific loss functions that introduces a controlled bias in the optimization process. These probabilities, informed by domain knowledge or empirical observations, help manage the exploration-exploitation trade-off by focusing optimization on certain areas while allowing exploration elsewhere.

In the domain of imbalanced regression, we conducted an investigation into the performance of different loss function schemes. Notably, we observed that employing a fixed mean squared error (MSE) loss for every other switch, while alternately switching between JSD and STD losses for the remaining switches, yielded results that were comparable to those obtained through a completely random search. This approach effectively reduced the exhaustive search space from a potentially large pool of $3^{switches}$ possible schemes to a significantly smaller set of schemes with a cardinality of $2^{\frac{switches}{2}}$, achieved by considering the binary switching pattern. By adopting this modified search strategy, we maintain strong performance in imbalanced regression tasks while significantly reducing the computational complexity. This focused exploration improves efficiency, enhances optimization, and aids in identifying near-optimal loss function configurations. We denote this method as SwitchLossR and Procedure 3 outlines the details.

Readers should note that restricted search techniques explore only a limited subset of possible optimization schemes. To maximize the chances of finding optimal results, conducting a more extensive search is recommended.

---

**Procedure 3** Restricted Search Space SwitchLoss for Imbalanced Regression

---

**Require:** $D$ - $\{(f_i, y_i)\}_{i=1}^{N}$ data set with $f_i$ feature vectors and $y_i$ continuous target values
$\qquad$ $\{MSE, JSD, STD_{loss}\}$ - set of loss functions
$\qquad$ *explores* - number of exploration cycles for the optimization scheme
$\qquad$ $\#switches$ - number of changes of loss function in traditional training
$\qquad$ *epochs* - number of epochs for the traditional optimization

$\quad$ **procedure** SWITCHLOSSR($D$)
$\qquad$ epochs_to_switch = round($\frac{epochs}{\#switches}$) $\qquad$ ▷ Number of epochs with a constant loss function
$\qquad$ switch_epochs = $\{$epochs_to_switch $\times$ i$\}_{i=1}^{\#switches-1}$
$\qquad$ loss_function = $MSE$ $\hfill$ ▷ Initialization
$\qquad$ $min\_error = \infty$ $\hfill$ ▷ Initialization
$\qquad$ best_model = Null $\hfill$ ▷ Initialization
$\qquad$ **for** $e \in (1, explores)$ **do** $\hfill$ ▷ First stage of exploration
$\qquad\qquad$ **for** $i \in (1, epochs)$ **do** $\qquad$ ▷ Nested traditional training within the exploration stage
$\qquad\qquad\qquad$ **if** $i \in switch\_epochs$ **then**
$\qquad\qquad\qquad\qquad$ **if** $switches$.index($i$)%2==0 **then**
$\qquad\qquad\qquad\qquad\qquad$ loss_function = $MSE$
$\qquad\qquad\qquad\qquad$ **else**
$\qquad\qquad\qquad\qquad\qquad$ loss_function = random($JSD, STD_{loss}$)
$\qquad\qquad\qquad\qquad$ **end if**
$\qquad\qquad\qquad$ **end if**
$\qquad\qquad\qquad$ traditional_training(D) $\hfill$ ▷ With previously defined loss_function
$\qquad\qquad$ **end for**
$\qquad\qquad$ Calculate error on a $balanced$ test data set $\hfill$ ▷ Compare optimization schemes
$\qquad\qquad$ **if** $error < min\_error$ **then** $\hfill$ ▷ Minimum error
$\qquad\qquad\qquad$ best_model = model $\hfill$ ▷ Save the model with the minimum test error
$\qquad\qquad\qquad$ $min\_error = error$
$\qquad\qquad$ **end if**
$\qquad$ **end for**
$\qquad$ **return** best_model $\hfill$ ▷ Return the model that performs the best
$\quad$ **end procedure**

---

## 4 EXPERIMENTAL EVALUATION

We designed an experimental setup targeted at assessing the performance of SwitchLoss in the context of imbalanced regression tasks.

### 4.1 DATA

For evaluating the performance of the presented approaches, we used three different types of datasets - 15 standard datasets from different imbalanced domains, 5 synthetic high-dimensional imbalanced datasets due to a special challenge that imblanace presents in high-dimensional settings, and two more complex age estimation image datasets IMDB-WIKI (Rothe et al., 2018) and AgeDB (Moschoglou et al., 2017), in order to evaluate the efficacy of our proposed method on deep learning architectures. Appendix contains tables that show in greater detail the main characteristics of these datasets and figures that show target value distributions, as well as the preprocessing that was done with image datasets. In total, we use 22 datasets that cover a range of sizes, feature numbers, distribution shapes and imbalance levels.

Important to note is that we split the data for the previous datasets into the training, validation, and test datasets. We selected the *balanced* validation and test datasets which implies that target values of the test and validation datasets are seeded uniformly throughout the whole target range. Random sampling from a data set would create imbalanced test data and consequently cause a bias towards a more abundant target value region in model performance assessment. Selected test data and validation data cover each 15% of the whole corresponding data set for standard and synthetic high-dimensional datasets.

Moreover, to ensure a comprehensive and unbiased evaluation of the SwitchLoss, we employ two distinct validation datasets. The first validation set is utilized during the traditional training stage to determine the optimal model within a specified range of epochs. In contrast, the second validation set is employed to identify the most effective optimization scheme. Subsequently, the selected optimization scheme is applied to train the model, and its performance is assessed using unseen data, thereby providing a robust evaluation of the proposed approach. Note that, the utilization of two validation sets consequently results in a smaller amount of training data used for SwitchLoss compared to other techniques.

### 4.1.1 IMBALANCED REGRESSION METHODS

We applied to each of the datasets, 4 different strategies. The techniques that we tested are as follows:

- Original data set with MSE loss
- SMOGN algorithm with MSE loss
- Original data set with generalized SwitchLoss (completely random search in exploration phase shown in Procedure 1)
- Original data set with restricted search space SwitchLoss, denoted as SwitchLossR (Procedure 3), with 32 exploration cycles

The two real image datasets are exceptions to this as, due to computational resource limitations, we do not apply the generalized SwitchLoss. Details of the parameters are given in the Appendix.

Based on our research and experimentation, we suggest switching loss functions 10 times. It is important to note that while this default value has demonstrated effectiveness across numerous datasets (as shown in the following section), it may not necessarily be the optimal choice for every data set.

To ensure fairness in our reporting, we adopt a single default parameter setting as the basis for presenting the main results. This approach is consistent with our treatment of SMOGN, where we employ only one setting despite the possibility of superior settings tailored to specific datasets.

### 4.1.2 LEARNING METHOD

We designed SwitchLoss to specifically address the optimization process of neural networks (NNs). A prerequisite for NNs to perform well is that the data is approximately balanced (Castro & Braga, 2013), (Wang et al., 2016). For standard and synthetic datasets we test 4 different architectures (deeper, shallower, wider and narrower) in order to show that performance is not architecture-specific: (16, 8, 4), (32, 16, 8), (40, 20, 10, 5), (64, 16, 4, 2). Listed architectures represent a number of hidden layers and the corresponding number of neurons per layer.
Furthermore, since the real-world image datasets IMDB-WIKI and AgeDB are more complex, we used deep ResNet architecture (He et al., 2016) for the learning process. Details of the implementation are provided in the Appendix.

### 4.1.3 EVALUATION METRICS

As proposed in (Liu et al., 2019) and adapted for regression in (Yang et al., 2021), we divide the target space into three disjoint subsets: many-shot region (bins with over 100 training samples), medium-shot region (bins with 20-100 training samples), and few-shot region (bins with under 20 training samples), and report results on these subsets, as well as overall performance. For metrics, we used root-mean-squared error (RMSE).

### 4.2 RESULTS

We present here an analysis of the performance of the different versions of the SwitchLoss algorithm on the datasets utilized in our experiments. The important findings are summarized in Table 1 for a combined overview including the generalized SwithLoss, and the restricted search SwitchLoss denoted as SwitchLossR to avoid confusion. Note that SwitchLoss and SwitchLossR approaches

Table 2: Evaluation of the performance for the image datasets.

| DataSet | Technique | Overall RMSE | Many | Medium | Few |
|---------|-----------|--------------|------|--------|-----|
| IMDB-WIKI | MSE | 138.06 | 108.70 | 366.09 | 964.92 |
| | SMOGN | 136.09 | 109.15 | 339.09 | 944.20 |
| | SwitchLossR | **132.59** | **106.87** | **328.68** | **886.79** |
| AgeDB | MSE | 101.60 | 78.40 | 138.52 | 253.74 |
| | SMOGN | 117.29 | 101.36 | 133.86 | **232.90** |
| | SwitchLossR | **99.60** | **77.36** | **125.54** | 240.13 |

can be combined, which is what the "Combined" column shows. In such a combined approach, we consider the algorithm as comprising 100 cycles of random search followed by 32 cycles within a region of the restricted search (or vice versa), and therefore 132 exploration cycles in total. This will be further discussed in the following section. For the sake of brevity, we solely report the overall root mean squared error (RMSE) winner for each data set.

Table 1: Number of best-performing datasets per technique and per neural network architecture. SwitchLoss and SwitchLossR against SMOGN and MSE.

| DataSetType | Architecture | MSE | SMOGN | SwitchLoss | SwitchLossR | Combined |
|-------------|--------------|-----|-------|------------|-------------|----------|
| standard | (16, 8, 4) | **6** | 3 | 3 | 3 | **6** |
| | (32, 16, 8) | 5 | 4 | 4 | 2 | **6** |
| | (40, 20, 10, 5) | 4 | 2 | 4 | 5 | **9** |
| | (64, 16, 4, 2) | 4 | **6** | 4 | 1 | 5 |
| | All | 19 | 15 | 15 | 11 | **26** |
| synth_HD | (16, 8, 4) | 0 | 2 | 2 | 1 | **3** |
| | (32, 16, 8) | 1 | **2** | **2** | 0 | **2** |
| | (40, 20, 10, 5) | **2** | **2** | 1 | 0 | 1 |
| | (64, 16, 4, 2) | 2 | 0 | 2 | 1 | **3** |
| | All | 5 | 6 | 7 | 2 | **9** |

Given the increased complexity of the age estimation image datasets in terms of size and neural network configuration, we provide detailed information regarding evaluation metrics across all target regions, as defined in subsection 4.1.3. The results of this evaluation are presented in Table 2.

We provide more comprehensive results in the Appendix, like separate findings for the two versions of the SwitchLoss algorithm to give a reader an idea of a tradeoff between the exploration space and results obtained with different levels of resource limitations.

## 5 DISCUSSION

In the previous section results of our experimental evaluation are presented. We have shown that generalized and restricted search SwitchLoss are comparable, while their combination outperforms the existing state-of-the-art approaches for imbalanced regression problems. Nevertheless, there are different aspects to discuss.

We presented individual and combined results for both generalized SwitchLoss and restricted search SwitchLossR algorithms. A combination of these algorithms leverages the strengths of both. As previously explained the combined approach consists of 132 exploration cycles (100 cycles of random search followed by 32 cycles within a region of the restricted search or vice versa). The two methods are exploring two different spaces. Both of them look for the best configuration of loss functions (among the space they search through). Combining them just expands the search space, making them additive. When looking for the minimum for each dataset we compare all 4 possibilities including SwitchLoss and SwitchLossR. If one of the best-performing methods is either SwitchLoss or SwitchLossR, that means that one of the best configurations is among the 132 possibilities ex-

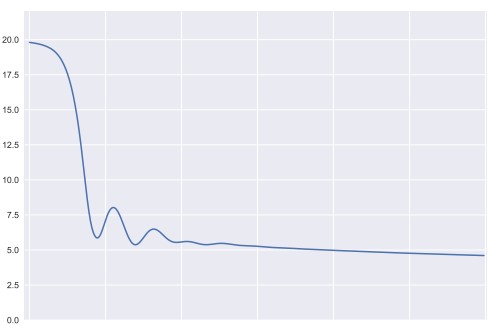

Figure 1: Convergence of validation error (Y axis) per epoch (X axis), for Accel data set, and (32,16,8) NN architecture, with MSE loss function.

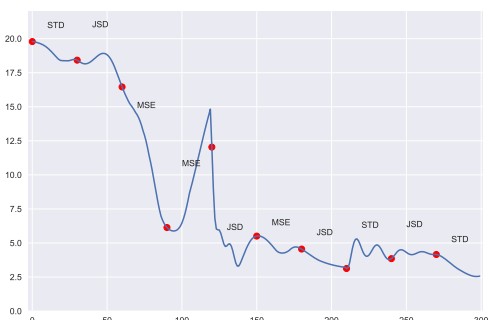

Figure 2: Convergence of validation error (Y axis) per epoch (X axis), for Accel data set, and (32,16,8) NN architecture, with SwitchLoss. Red dots represent switching epochs.

plored by the combination of methods. By integrating these two strategies, the aim is to achieve a balance between exploration and exploitation, harnessing the advantages of both approaches. The randomized nature of the generalized search helps in avoiding local optima and discovering diverse regions of the search space, while the restricted search focuses on refining the solutions within a specific region.

Figures 1 and 2 provide a visual representation of the convergence of validation error per epoch, for Accel data set and (32,16,8) NN architecture. An important observation is that optimization with mean squared error (MSE) exhibits greater stability during training. However, despite its relative instability, optimization with SwitchLoss achieves a validation error 50% smaller than that of MSE. The inherent instability associated with the switch of loss functions is a direct consequence of utilizing multiple functions instead of a single one. While this characteristic may lead to fluctuations in the optimization process, it also serves as a mechanism to prevent the optimization from becoming trapped in local minima, as commonly experienced with MSE in imbalanced regression problems. By employing the switch of loss functions, the optimization process gains the ability to explore a wider range of solution spaces, thereby increasing the likelihood of finding more favorable minima. Consequently, although the convergence may not exhibit the same level of stability as MSE, the resulting validation error achieved through SwitchLoss is significantly reduced. It is important to highlight that we do not extend the training duration for the SwitchLoss experiments. The same number of epochs is used as in the original data and MSE loss experiments. This deliberate choice ensures that SwitchLoss does not gain any undue advantage, maintaining a fair comparison across all experiments.

Another substantial aspect of this research is the comparison between the generalized SwitchLoss and the restricted search space variant, SwitchLossR. Our analysis shows that SwitchLoss wins in 55% of standard dataset cases and 75% in the high-dimensional domain, benefiting from exploring a wider range of optimization schemes. However, SwitchLossR performs better in nearly half of the standard datasets, as its more focused approach allows it to explore specific areas more efficiently, leading to quicker and often better results in certain cases. Although the exploration space of SwitchLoss is three times larger than that of SwitchLossR, it remains limited in comparison to the vast array of all conceivable possibilities. This implies that with the larger exploration stage, SwitchLoss is more likely to encompass some of the schemes depicted by SwitchLossR. Nonetheless, it is worth noting that the exploration space of these two algorithms could but does not necessarily overlap.

The speed of execution is a relevant consideration when evaluating an algorithm such as SwitchLoss. In our study, we have deliberately chosen to report results based on only 100 exploration cycles, out of a total of 59,049 possible schemes. This decision is made to ensure that the computational complexity of the algorithm remains manageable, without demanding extensive resources in terms of time or computational power.

It is worth noting that the speed of execution for SwitchLoss follows a time complexity of $O(e)$, where $e$ represents the number of exploration cycles. In summary, while acknowledging the potential benefits of a more exhaustive exploration stage, our study demonstrates that even a limited number of exploration cycles can lead to competitive performance compared to alternative techniques.

Table 2 shows an important feature of SwitchLoss. It does not only improve overall errors but also errors in distinct target regions. It can be noted that SMOGN worsens "many" shots region for IMDB-WIKI (Rothe et al., 2018) despite improving the overall RMSE. Depending on the problem, some regions can be more valuable for prediction than others. Our research shows that deep architectures are better able to leverage the benefits offered by SwitchLoss. Furthermore, when more data is available, heuristics such as Jensen-Shannon divergence and standard deviation exhibit greater precision, resulting in amplified advantages derived from SwitchLoss. The level of class imbalance also plays a role, as we find that for less skewed distributions, the regular mean-squared error more frequently outperforms SwitchLoss. Conversely, in highly imbalanced cases, SwitchLoss contributes more significantly.

Furthermore, the work by Blagus and Lusa (2013) suggests that SMOTE-based techniques introduce bias and perform worse than baseline methods in high-dimensional settings. Our experiments align with these findings, as we observe that SMOGN under-performs on image datasets in comparison to standard datasets.

We did not assess the combined usage of SMOGN and SwitchLoss in our research, even though there are no technical barriers to combining them. SMOGN addresses data, while SwitchLoss focuses on optimization. However, the core concept of SwitchLoss aims to prevent under- or over-sampling, rendering its combination with SMOGN inconsequential.

In summary, our comparative analysis demonstrates that SwitchLoss exhibits superior performance on datasets with ample samples, high imbalance, and complex NN architectures.

# 6 CONCLUSION

In conclusion, this research paper presents the novel optimization scheme SwitchLoss which represents a versatile approach to the optimization procedure for neural networks. While the applicability of this scheme is not limited by any problem type, the specific focus of this paper is directed towards addressing the challenges associated with imbalanced regression.

The SwitchLoss approach entails a nested framework that combines the exploration of diverse loss function schemes with the conventional training methodology employed in neural network optimization. It comprises two stages: the exploration stage and the traditional neural network training stage. In the exploration stage, various loss function schemes are investigated, with a predefined number of cycles. The goal is to assess the impact of different loss function configurations on the optimization process and identify the most effective scheme for improving model performance. In the subsequent traditional training stage, the neural network model is trained using the optimized loss function scheme obtained from the exploration stage.

This paper makes significant contributions in several key areas. Firstly, it introduces the optimization scheme SwitchLoss, which is specifically tailored to address the challenges of imbalanced regression. Additionally, a variant of SwitchLoss, called SwitchLossR, is presented as a means to reduce computational complexity while maintaining effectiveness. The effectiveness of both SwitchLoss and SwitchLossR is thoroughly evaluated on 15 standard and 5 synthetic high-dimensional datasets, representing diverse data distributions. The results demonstrate that the combination of SwitchLoss and SwitchLossR outperforms other existing techniques. While SwitchLoss generally performs better than SwitchLossR, particularly in the high-dimensional domain, it is noteworthy that SwitchLossR surpasses SwitchLoss for nearly half of the standard datasets despite its smaller search space. Furthermore, experiments conducted on more complex age estimation image datasets, specifically AgeDB (Moschoglou et al., 2017) and IMDB-WIKI (Rothe et al., 2018), highlight the superior performance of SwitchLossR compared to other techniques within the context of deep learning architectures. Finally, we observe that the SwitchLoss schemes exhibit superior performance on datasets with abundant samples, high imbalance, and complex neural network architectures.

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

# A APPENDIX

## A.1 DATA

Table 3 shows characteristics of standard datasets. N represents the number of samples in a data set, f.total is the number of features, f.nom is the number of nominal features and f.num is the number of numeric predictors. nRare is the number of samples with relevance value (determined by Ribeiro (2011)) higher than the threshold (0.8) and finally %Rare represents a percent of rare samples compared to the entire data set size. Figure 3 shows target value distributions for each of the 15 standard datasets.

Table 3: Standard datasets information.

| DataSet | N | f.total | f.nom | f.num | nRare | %Rare |
|---------|------|---------|-------|-------|-------|-------|
| Abalone | 4177 | 8 | 1 | 7 | 679 | 16.3 |
| Accel | 1732 | 15 | 3 | 12 | 89 | 5.1 |
| a1 | 198 | 11 | 3 | 8 | 28 | 14.1 |
| a2 | 198 | 11 | 3 | 8 | 22 | 11.1 |
| a3 | 198 | 11 | 3 | 8 | 32 | 16.2 |
| a4 | 198 | 11 | 3 | 8 | 31 | 15.7 |
| a6 | 198 | 11 | 3 | 8 | 33 | 16.7 |
| a7 | 198 | 11 | 3 | 8 | 27 | 13.6 |
| availPwr | 1802 | 16 | 7 | 9 | 157 | 8.7 |
| bank8FM | 4499 | 9 | 0 | 9 | 288 | 6.4 |
| boston | 506 | 13 | 0 | 13 | 65 | 12.8 |
| cpuSm | 8192 | 13 | 0 | 13 | 713 | 8.7 |
| fuelCons | 1764 | 38 | 12 | 26 | 164 | 9.3 |
| heat | 7400 | 11 | 3 | 8 | 664 | 8.9 |
| maxTorque | 1802 | 33 | 13 | 20 | 129 | 7.2 |

Table 4 shows details of synthetic high-dimensional datasets. We use two different methods for generating the synthetic data. In the first method, the target value is generated by applying a random linear regression model to the previously generated input and a Gaussian-centered noise with

Table 4: Synthetic high-dimensional datasets information.

| DataSet | N | f.total | nRare | %Rare | Method |
|---|---|---|---|---|---|
| synthHD_1 | 293 | 1000 | 82 | 27.9 | make_regression |
| synthHD_2 | 2228 | 6000 | 89 | 23.3 | make_regression |
| synthHD_3 | 500 | 20000 | 44 | 8.8 | MLP |
| synthHD_4 | 300 | 15000 | 22 | 7.3 | MLP |
| synthHD_5 | 700 | 15000 | 37 | 5.2 | MLP |

Table 5: Image datasets information.

| DataSet | N | im.dim | nRare | %Rare | test.size | val.size |
|---|---|---|---|---|---|---|
| IMDB-WIKI | 213553 | 224 × 224 | 17315 | 8.1 | 11022 | 11022 |
| AgeDB | 16488 | 224 × 224 | 293 | 1.8 | 2140 | 2140 |

an adjustable scale (make_regression) (Pedregosa et al., 2011). We also resort to a Multilayer Perceptron (MLP) as a random function to generate the remaining synthetic datasets. This assumes that the function can be learned again by an MLP. Our network's parameters are initialized with a standard Gaussian distribution. The features are also drawn from a standard Gaussian distribution. The network consists of 3 hidden layers (30, 10, 3 neurons per layer, respectively) and ReLU (Nair & Hinton, 2010) activation. The final hidden layer is connected to a single neuron with linear activation to obtain target values for a regression task. We designed the datasets to cover a wide range of sample and feature sizes, their ratios, the percentage of rare data and to have present one or two extremes. Figure 4 shows target value distributions for 5 synthetic datasets.

Table 5 shows the main features of image age estimation datasets. im.dim represents a dimension of images once processed. Since the sizes of these datasets are more significant than the previous ones columns test.size and val.size show the corresponding test/validation number of samples. Figure 5 show the age distribution in these datasets. The test and validation data is *balanced* as well.

## IMDB-WIKI

The IMDB-WIKI dataset (Rothe et al., 2018) is a large face image dataset for age estimation from a single input image. The original version contains 523.0K face images and the corresponding ages, where 460.7K face images are collected from the IMDB website and 62.3K images from the Wikipedia website. We construct IMDB-WIKI by first filtering out unqualified images with low face scores (Rothe et al., 2018), and then manually creating balanced validation and test set over the supported ages. Overall, the curated dataset has 191.5K images for training, and 11.0K images for validation and testing, respectively. We make the length of each bin to be 1 year, with a minimum age of 0 and a maximum age of 186. The number of images per bin varies between 1 and 7,149, exhibiting significant data imbalance. As for the data pre-processing, the images are first resized to 224 × 224. During training, we follow the standard data augmentation scheme (He et al., 2016) to do zero-padding with 16 pixels on each side, and then random crop back to the original image size. We then randomly flip the images horizontally and normalize them into [0, 1].

## AGEDB

The original AgeDB dataset (Moschoglou et al., 2017) is a manually collected in-the-wild age database with accurate and noise-free labels. Similar to IMDB-WIKI, the task is also to estimate age from visual appearance. The original dataset contains 16,488 images in total. We construct AgeDB in a similar manner as IMDB-WIKI, where the training set contains 12,208 images, with a minimum age of 0 and a maximum age of 101, and maximum bin density of 353 images and minimum bin density of 1. The validation set and test set are made balanced with 2,140 images. Similarly, the images in AgeDB are resized to 224 × 224, and go through the same data pre-processing schedule as in the IMDB-WIKI dataset.

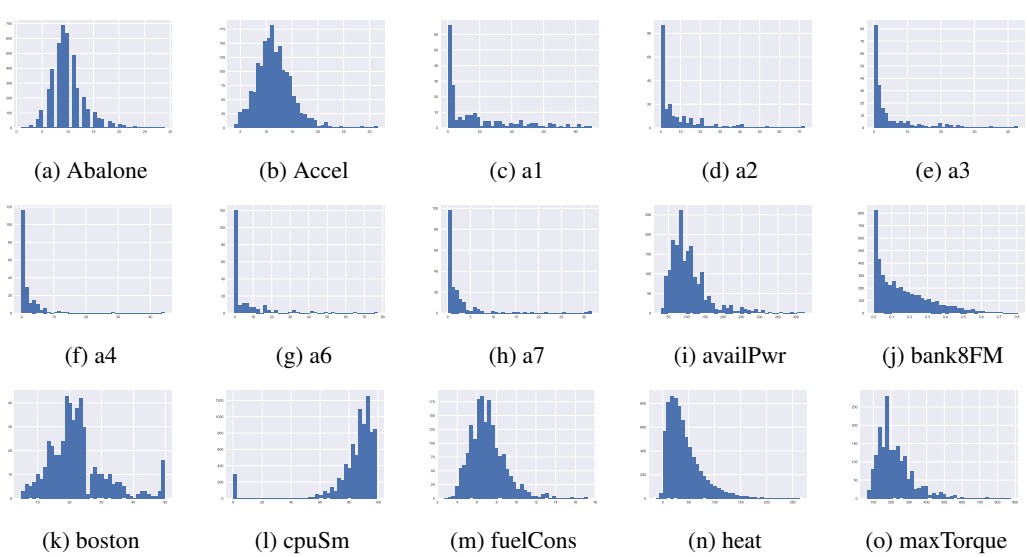

Figure 3: Distributions of target values of standard datasets.

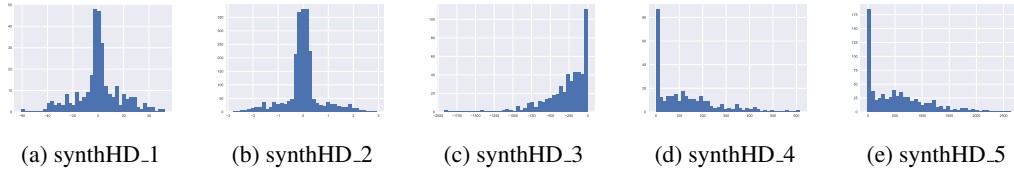

Figure 4: Distributions of target values of HD datasets.

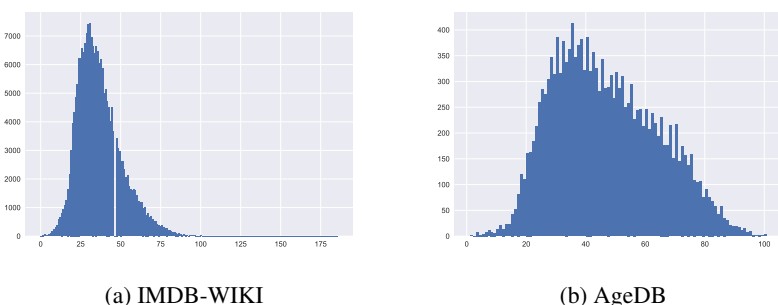

Figure 5: Distributions of target values for image datasets.

## A.2 PARAMETERS

Default values for SMOGN are used (Kunz, 2020): $k = 5$ specifies the number of neighbors to consider for interpolation in over-sampling, $pert = 0.02$ represents the amount of perturbation to apply to the introduction of Gaussian Noise, balanced sampling is selected, replacement is not selected in under-sampling and relevance function threshold is set to be 0.8 (as in the original paper (Branco et al., 2017)).

Training is run for 300 epochs, we use Adam optimization (Kingma & Ba, 2014), and a learning rate of $10^{-2}$. These specific values have been shown to cause convergence of all models for all datasets.

### RESNET

We use the ResNet-50 model (He et al., 2016) for all IMDB-WIKI and AgeDB experiments. We train all models for 90 epochs using the Adam optimizer (Kingma & Ba, 2014), with an initial learning rate of $10^{-3}$ and then decayed by 0.1 at the 60-th and 80-th epoch, respectively. We fix the batch size as 256.

## A.3 ADDITIONAL RESULTS

The values depicted in the pie charts represent the number of datasets associated with a particular strategy that exhibits the best overall performance within the respective data set type. Figure 6 illustrates comparison of all methods for standard datasets, while Figure 7 shows that for high-dimensional datasets. Figure 8 and 9 separate the findings for the generalized SwithLoss against SMOGN and MSE, while Figure 10 and Figure 11 represent findings for the restricted search space SwitchLossR against SMOGN and MSE. It is worth noting that the reported numbers are aggregated across all four neural network architectures.

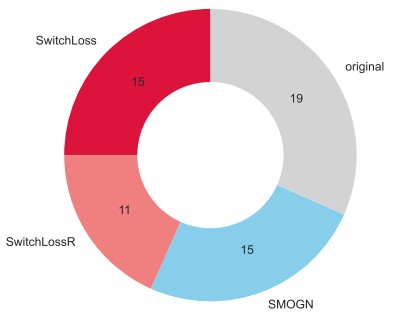

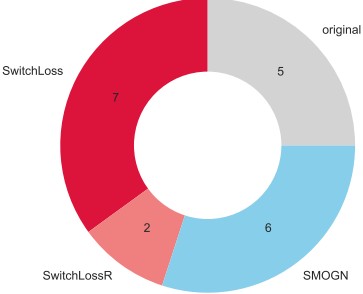

Figure 6: Pie chart of best-performing strategies for standard data. Testing generalized SwitchLoss and restricted search SwitchLossR against SMOGN and original MSE.

Figure 7: Pie chart of best-performing strategies for synthetic HD data. Testing generalized SwitchLoss and restricted search SwitchLossR against SMOGN and original MSE.

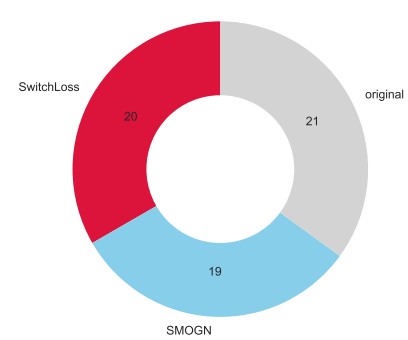

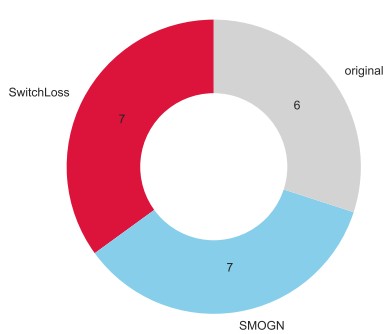

Figure 8: Pie chart of best-performing strategies for standard data. Testing generalized SwitchLoss against SMOGN and original MSE.

Figure 9: Pie chart of best-performing strategies for synthetic HD. Testing generalized SwitchLoss against SMOGN and original MSE.

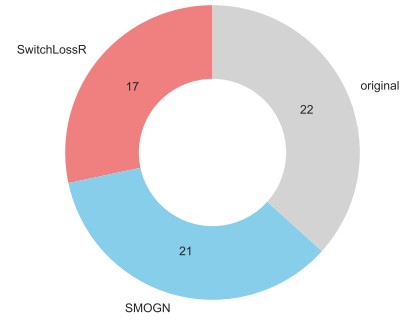

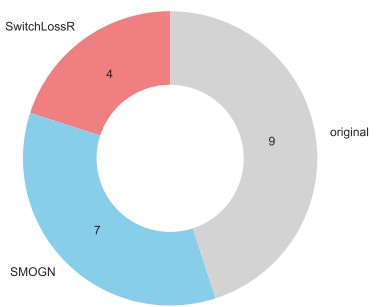

Figure 10: Pie chart of best-performing strategies for standard data. Testing restricted search space SwitchLossR against SMOGN and original MSE.

Figure 11: Pie chart of best-performing strategies for synthetic HD. Testing restricted search space SwitchLossR against SMOGN and original MSE.

Figures 12 and 13 show a direct comparison between SwitchLoss and SwitchLossR in the best-performing number of datasets.

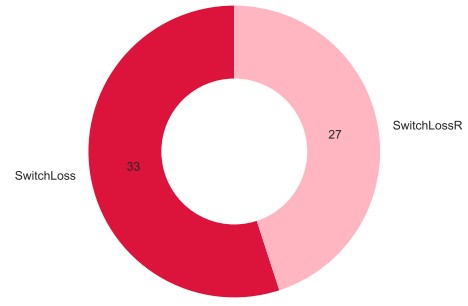

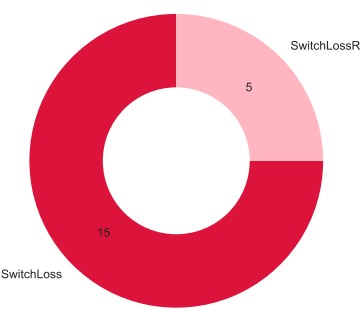

Figure 12: Pie chart of best-performing strategies for standard data. Testing generalized SwitchLoss against restricted search space SwitchLossR.

Figure 13: Pie chart of best-performing strategies for synthetic HD. Testing generalized SwitchLoss against restricted search space SwitchLossR.

Table 6 compares generalized SwitchLoss with 100 exploration epochs against other techniques, while Table 7 compares restricted search SwitchLoss, denoted as SwitchLossR, against other techniques.

Table 6: Number of best-performing datasets per technique and per neural network architecture. SwitchLoss against SMOGN and MSE.

| DataSetType | Architecture | MSE | SMOGN | SwitchLoss |
|---|---|---|---|---|
| standard | (16, 8, 4) | 7 | 3 | 5 |
| | (32, 16, 8) | 5 | 6 | 4 |
| | (40, 20, 10, 5) | 5 | 4 | 6 |
| | (64, 16, 4, 2) | 4 | 5 | 6 |
| synth_HD | (16, 8, 4) | 1 | 2 | 2 |
| | (32, 16, 8) | 1 | 2 | 2 |
| | (40, 20, 10, 5) | 2 | 2 | 1 |
| | (64, 16, 4, 2) | 3 | 0 | 2 |

Table 7: Number of best-performing datasets per technique and per neural network architecture. SwitchLossR against SMOGN and MSE.

| DataSetType | Architecture | MSE | SMOGN | SwitchLossR |
|---|---|---|---|---|
| standard | (16, 8, 4) | 7 | 3 | 5 |
| | (32, 16, 8) | 5 | 6 | 4 |
| | (40, 20, 10, 5) | 5 | 4 | 6 |
| | (64, 16, 4, 2) | 4 | 5 | 6 |
| synth_HD | (16, 8, 4) | 1 | 2 | 2 |
| | (32, 16, 8) | 1 | 2 | 2 |
| | (40, 20, 10, 5) | 2 | 2 | 1 |
| | (64, 16, 4, 2) | 3 | 0 | 2 |

## A.4 ADDITIONAL DISCUSSION

### JENSON-SHANNON DIVERGENCE VS KULLBACK-LEIBLER DIVERGENCE

Formally, for two probability distributions p and q, the Jensen-Shannon divergence (JSD) is defined as the average of the Kullback-Leibler (KL) divergences $D_{KL}(p\|q) = \sum_x^N p(x)log(\frac{p(x)}{q(x)})$ between p and the average distribution obtained by mixing p and q noted as $m = \frac{1}{2}(p + q)$, and between q and m. One advantage of JSD with respect to KL divergence is its symmetric nature. The asymmetry can lead to different optimization behaviors and potentially biased results. Moreover, JSD has a bounded range, with values between 0 and 1, making it more interpretable and easier to compare across different contexts (Thiagarajan & Ghosh, 2023). KL divergence, on the other hand, is unbounded and can take on large values, potentially leading to numerical instability and difficulties in optimization. Another advantage of JSD is its robustness in situations where the two distributions being compared have overlapping support. Unlike KL divergence, which can be sensitive to regions with zero probability in one of the distributions, JSD can handle such cases effectively (Thiagarajan & Ghosh, 2023).

### RECOMMENDATION

While it is evident that optimal results are more likely to be achieved with a comprehensive exploration of all possibilities, we demonstrate that in many cases, even a limited number of exploration cycles, such as 100 or 32 in the case of restricted search, can yield superior performance compared to alternative techniques. However, it is important to note that if sufficient resources are available, we recommend a broader exploration phase to further enhance the algorithm's effectiveness.

