# OpenReview forum: "SwitchLoss: A Novel Optimization Scheme for Imbalanced Regression"
_ICLR.cc/2025/Conference — ICLR 2025 Conference Withdrawn Submission_

### Official Review · Reviewer_i2kQ · 2024-11-01

**Soundness:** 1
**Presentation:** 1
**Contribution:** 2
**Rating:** 3
**Confidence:** 4

**Summary:**

The authors introduce SwitchLoss, an optimization framework in which loss function schemes are first selected in an exploration phase and then used in a training phase for neural network optimization.
The key idea is to alternate between several loss functions during optimization.
The proposed method and a restricted variation thereof are evaluated on imbalanced regression tasks ranging from standard tabular benchmark data to age estimation.

**Strengths:**

- The paper is concerned with the interesting and practically important topic of imbalanced regression

**Weaknesses:**

- The novelty of the proposed general SwitchLoss framework is limited. Consider e.g. (Sculley, 2010) who uses such an approach to optimize a combined loss of regression and ranking terms.

- I am highly doubtful regarding the experimental results. Consider the results given in Table 1. If I see it correctly, this table presents the results on the 15 standard datasets. First of all, the results do not coincide with the results in Appendix (see Table 6 and Table 7). Secondly, the indiidual rows do not sum up to 15. The first 4 itmes of each row do, while the "Combined" column is the sum of the SwitchLoss and SwitchLossR column. It is explained in the manuscript text, that this is the case because it harnesses the advantages of both and thus should be additive. To me, it did not become clear whether the experiments were actually run for a combined implementation or whether the other two columns where simply summed up. Also, I think this can not hold in general, as the exploration and training is done on train data. For the validation performance, I doubt that the number of winners for a combined approach is always the sum of the two others, because one approach may be better suited on the training data while another is advantageous on test data.
Additionally reporting only the number of best performing methods hides the margin between the methods performances. I understand that these numbers cannot be trivially aggregated for presentation, but I would like to see some RMSE values achieved by the methods at least in the appendix.

- I also question the experimental setup and choice of datasets. In many cases MSE outperforms SMOGN. Why is such a standard regression loss better suited than the SOTA for imbalanced regression?

- The presentation of the proposed method is highly redundant. The manuscript has 3 pseudocodes which are almost identical.

Literature:
Sculley, D. (2010, July). Combined regression and ranking. In Proceedings of the 16th ACM SIGKDD international conference on Knowledge discovery and data mining (pp. 979-988).

Minor remarks:
- p. 7 line 377 "SwithLoss" -> "SwitchLoss"
- p. 8 Table 2 appears before Table 1

**Questions:**

- Is the combined version of SwitchLoss and SwitchLossR implemented and evaluated or are the results for the runs of the individual versions summed up?

- Why does MSE show such a competitive performance to SMOGN?

---

### Official Review · Reviewer_5HJB · 2024-11-02

**Soundness:** 3
**Presentation:** 2
**Contribution:** 2
**Rating:** 3
**Confidence:** 4

**Summary:**

The submission presents an approach to tackle deep regression problems that exhibit an imbalanced target variable. The basic idea is to switch between different loss functions during training; the submission considers MSE, KL-divergence, and a loss function that encourages the variance in the predicted target values to be similar to the variance in the ground truth. Given a grid on the number of epochs performed during training, the loss is switched randomly when an epoch in the grid is encountered.  This training process, with randomly switched loss functions, is repeated a certain number of times, as determined by a hyperparameter, and the model with the best performance, measured on validation data, is kept. The submission also has a modified version of this approach, where MSE is selected at every second grid point, and one of the other two losses is chosen randomly at the other grid points. In the submission's experiments with several configurations of fairly shallow multilayer perceptrons, a hybrid of the two approaches has a positive win/loss ratio when compared against a competing method for imbalanced regression called SMOGN, which is based on sampling. This hybrid also yields lower MSE than SMOGN on two age estimation datasets when ResNets are used.

**Strengths:**

The idea of switching between loss functions during training appears novel, at least in the context of imbalanced regression. The proposed method is simple and appears to be a valid competitor when compared to sampling-based imbalanced regression.

**Weaknesses:**

The submission does not present a strong theoretical justification for the proposed method.

There is no comparison to the DenseLoss method proposed by Steininger et al. that is cited in the paper.

The comparison to SMOGN does not seem entirely fair: the sequence of loss function can be viewed as a hyperparameter, which is optimized using random search on the validation data. Similarly, the hyperparameters of SMOGN should be optimized on validation data. The submission compares to SMOGN with default parameters instead.

The majority of the results are presented in terms of win/loss statistics. It is unclear how large the improvements in accuracy were.

Some important details seem to be missing:

a) KL-divergence requires probability estimates, and it is not stated how they were obtained.

b) It is not stated how the bins for the target variable were created.

c) It is not specified how the validation and the test data were balanced

There is a lot of redundancy in the pseudo code presented in the paper. Showing Procedure 2, a specialized version of Procedure 1, is not a good use of space. Similarly, Procedure 3 involves a minor modification of Procedure 2.

**Questions:**

N/A

---

### Official Review · Reviewer_QTc8 · 2024-11-08

**Soundness:** 2
**Presentation:** 2
**Contribution:** 2
**Rating:** 3
**Confidence:** 3

**Summary:**

This paper investigate the possibility of addressing imbalanced data in regression problems using various loss function. In particular, this paper suggests to use thee different loss functions alternatively during the training process and shows empirically the performance can be improved (in terms of overall MSE and MSE for disjoint subsets of target space).

**Strengths:**

The paper proposes a simple method to handle imbalanced dataset for regression tasks. The method is straight forward and easy to implement.

**Weaknesses:**

1. The paper does not provide a rigorous  definition of imbalanced regression problem. There is not mathematical definition for imbalanced dataset. Does imbalanced data mean that the distribution of $Y$ has long tail and we do not see enough sample from the long tail? Is it possible that we have very few sample for a specific bin even if that bin has high probability?

2. There is no clear definition of balanced test data. In the proposed algorithm, we need to have access to balanced test data set. How this balanced test data is generated? I am assuming that we need to split the target space to several bins and we should make sure that we have the same number of samples in each bin. However, it is not clear how we should split the target space. Depending how we define bins, the balanced dataset will be different.

3. The experiment part is very limited. This paper is purely empirical paper. So in order to evaluate the algorithm, we need to see more experiments. For example, I did not find Table 2 useful. Because the results highly depend on how we define the bins for target space. I believe we can select the bins such that the proposed method looks better in terms of MSE in each bin. Should we discuss the impact of bins on the results? I believe more experiment is needed to understand the performance of the proposed under different target space split.

4. The authors compare their method only with one baseline that can handle imbalanced data. Is there any other baseline that we need to consider for imbalanced regression? Is there any method for imbalanced classification that can be extended to regression?

5. The numbers in the tables are not reliable. In particular, there is no variance reported in the tables.

**Questions:**

I want to ask authors to address the weaknesses that I pointed out above.

---

### Note · Authors · 2024-11-14

I have read and agree with the venue's withdrawal policy on behalf of myself and my co-authors.